**communications** engineering

# A compact X-ray source via fast microparticle streams
Rolf Behling [1,2] ✉, Christopher Hulme [3], Gavin Poludniowski[2,4], Panagiotis Tolias[5] & Mats Danielsson[1]

The spatiotemporal resolution of diagnostic X-ray images is limited by the erosion and rupture of conventional stationary and rotating anodes of X-ray tubes from extreme density of input power and thermal cycling of the anode material. Conversely, detector technology has developed rapidly. Finer detector pixels demand improved output from brilliant keV-type X-ray sources with smaller X-ray focal spots than today and would be available to improve the efficacy of medical imaging. In addition, novel cancer therapy demands for greatly improved output from X-ray sources. However, since its advent in 1929, the technology of high-output compact X-ray tubes has relied upon focused electrons hitting a spinning rigid rotating anode; a technology that, despite of substantial investment in material technology, has become the primary bottleneck of further improvement. In the current study, an alternative target concept employing a stream of fast discrete metallic microparticles that intersect with the electron beam is explored by simulations that cover the most critical uncertainties. The concept is expected to have far-reaching impact in diagnostic imaging, radiation cancer therapy and non-destructive testing. We outline technical implementations that may become the basis of future X-ray source developments based on the suggested paradigm shift.

Medical diagnostic X-rays emerge primarily as bremsstrahlung by the interaction of a focused beam of electrons of up to 150 keV with atomic nuclei, preferably of tungsten. The energy conversion efficiency is of the order of 0.05%, taking the loss of photons by apertures and X-ray filters into account. Due to this inefficiency, an input power of the order of 100 kW is required to generate the necessary X-ray intensity of a few dozen Watts for computed tomography (CT) scanning. To cope with this intense and focused heat input, contemporary diagnostic X-ray tubes, Fig. 1a, comprise sintered rotating anode bodies. Figure 1b, titanium–zirconium–molybdenum composite discs of the highest available thermomechanical strength, often slotted and backed with carbon cylinders for heat storage, are covered with sub-millimetre tungsten–rhenium alloy X-ray conversion layers. This layer absorbs the incident electrons, serves to thermally shield the molybdenum-based bulk rotor member beneath it and generates useful X-rays. Rotating anodes must withstand bulk temperatures of 1600 °C, thermal gradients that stress the material up to its yield strength, high-frequency cycling with surface temperatures between 1600 °C and 2800 °C or more, and extreme centrifugal stress. Rotation provides local convection cooling of the interaction region, the X-ray focal spot. For safety reasons, the focal track velocity must be limited[1] to about 100 m s$^{-1}$, given the best available metallic anode material. The X-ray conversion layer, of the order of micrometres thick, that interacts with the electrons, undergoes up to $10^8$ thermal cycles during the service life that often ends with substantial cracking and erosion. Despite complex advances via the addition of hafnium carbide or molybdenum–niobium oxides, the progress of advanced processing and coating methods has recently faded. Surface-heating models are adequate to express the focal spot power density as a function of the rotor velocity[2,3], predicting that doubling the focal spot input power of a premium performance rotating anode tube would require an unbearable 16-fold increase of the centrifugal stress alone that adds to the thermomechanical stress. Faced with these limitations, the technological progress of sources has largely stalled.

After the introduction of finely pixelated detectors[4–7], Fig. 1d, attention switched to improving the technology relating to focal spot size reduction to enhance the resolution in medical imaging systems. This would allow for reducing artefacts and image noise as well as improving diagnostic efficacy for attenuation imaging[8–11], with dark-field X-ray imaging[12–14], and FLASH, mini-beam, and micro-beam clinical orthovolt radiation therapy[15] seeing

[1]Particle Physics, Astrophysics and Medical Imaging Department, KTH Royal Institute of Technology, Stockholm, Sweden. [2]Department of Clinical Science, Intervention and Technology, Karolinska Institutet, Stockholm, Sweden. [3]Department of Materials Science and Engineering, KTH Royal Institute of Technology, Stockholm, Sweden. [4]Department of Nuclear Medicine and Medical Physics, Karolinska University Hospital, Stockholm, Sweden. [5]Department of Space and Plasma Physics, KTH Royal Institute of Technology, Stockholm, Sweden. ✉e-mail: Rkobe@kth.se

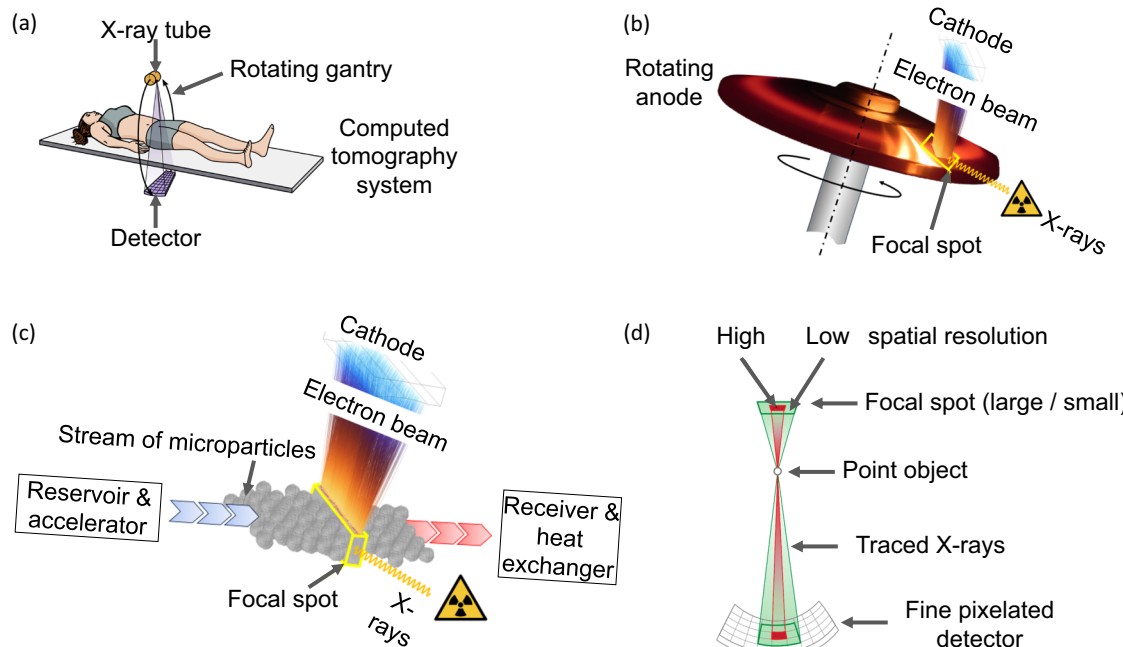

**Fig. 1 | Components and key aspects of X-ray imaging. a** Typical system configuration for computed tomography. **b** Working principle of the state-of-the-art rotating anode. **c** Following the ideas presented in this paper, the rotating anode can beneficially be replaced by a microparticle stream, enabling a substantially reduced X-ray focal spot with X-ray output and resolution much higher than is possible today. **d** Basic imaging geometry with an X-ray source, a sample point object, and a detector. If the focal spot size is shown significantly larger than the pixel size, which is the case today for medical computed tomography, the image will be blurred (wide green beam), while with the microparticle stream source (red) the image quality is far less compromised.

particular benefits. Currently, a premium tier X-ray tube permits for a standardised CT anode input power of 120 kW that can be applied for a 4 s CT scan every 10 min with a standardised focal spot size of about 1.1 nominal[16], at a target angle of 8°. This translates to a maximum physical focal spot width of 1.5 mm and a real (non-projected) length on the rotating target of 15.8 mm that defines the circular focal spot track. However, given the restrictions of power density, a smaller and highly appreciated nominal focal spot of 0.3 would only allow for the use of about 25 kW electronic input power, too small to gather enough photons in each detector pixel in high-speed CT. Indeed, highly brilliant X-radiation is delivered by other technology, for example, Thomson scatter[17], synchrotron[18–21], or laser plasma sources[22], or from liquid metal jet anodes[23]. However, their photon flux is insufficient for medical imaging. While cathodes could deliver more electron current[24], the erosion of the target remains the limiting factor.

We show that one way to circumvent these drawbacks is replacing the rigid anode with a stream of microparticles, Fig. 1c, without changing the fundamental mechanism of bremsstrahlung generation. Tungsten microparticles of 1 μm diameter and larger are commercially available for several applications. They may be mechanically and/or electrically accelerated in a vacuum, e.g., with magnetically levitated spinning members that are free of thermal stress and allow for high rotor velocities, which may receive the microparticles close to their centre and transfer the necessary momentum by centrifugal force to channel and expel them radially as a thin stream. Such magnetic-borne rotor systems are the mature technology for ultra-high vacuum turbo-molecular pumps.

After interaction with the electron beam, the microparticles may be received by stationary or rotating cooling members in a distance that gradually slows them down and conveys them back into a reservoir that may be backfilled by gravity after each CT scan when the tube stops in a suitable position of the otherwise rotating CT gantry. Alternatively, we suggest the more complex use of gradients of centrifugal acceleration during rotation of the CT gantry for such a purpose.

In contrast to conventional anodes that comprise oxides and carbides for high yield strength and produce high gas load when operated as indicated in Fig. 1b, highly brilliant gas-sensitive cathodes[1] may be employed

with the suggested pure tungsten microparticles, enabled by substantially reducing the residual gas pressure of the dominating carbon monoxide. As another advantage, the expensive rhenium additive would become obsolete.

Additional flexibility for design and engineering of the X-ray source arises from the spatial decoupling of heating and cooling of the target material in contrast to the focal spot of conventional rotating anodes that serves both functions in a micrometre thin superficial volume.

Several issues may arise from the use of microparticles. Typically, dust and microparticles are undesired in electronics manufacturing[25], nuclear fusion devices[26] and X-ray sources[27–29]. To avoid pollution of the region of high voltage insulation, any compact electron source might need to be separated from the particle-contaminated target space. Fortunately, both spaces are under high vacuum. This would enable the use of very thin electron transparent carbon or diamond windows or other strong materials. Alternatively, microparticles may be actively charged after interaction and kept inside the target space with electrically biased repelling electrodes.

In summary, we are convinced that development according to the new paradigm will enable advanced manipulation of fast-flowing tungsten microparticles, Fig. 1c, and enable the improvement of X-ray imaging. The current work demonstrates that the most critical aspect of charging microparticles can be managed. The traditional paradigm that the anode was necessary to be conductively connected to the power source can be abandoned.

In the following description of the concept, we will firstly discuss the switch from surface heating of a rigid body to volume heating of a stream of discrete microparticles[2,3,30,31] before Finally, the gain of permitted input power density and image resolution is predicted, and other benefits are listed.

Figure 2 sketches a stream of microparticles interacting with an orthogonal parallel beam of electrons. Microparticles enter from the left and exit the interaction region heated to the right. For a wide stream, back-scattered electrons may enlarge the interaction region beyond the cross-section of the primary electron beam that typically defines the focal spot of a rigid anode. To minimise X-ray coronae, the breadth, $B$, and/or the thickness, $H$, of the microparticle stream may be minimised. The reduction of the

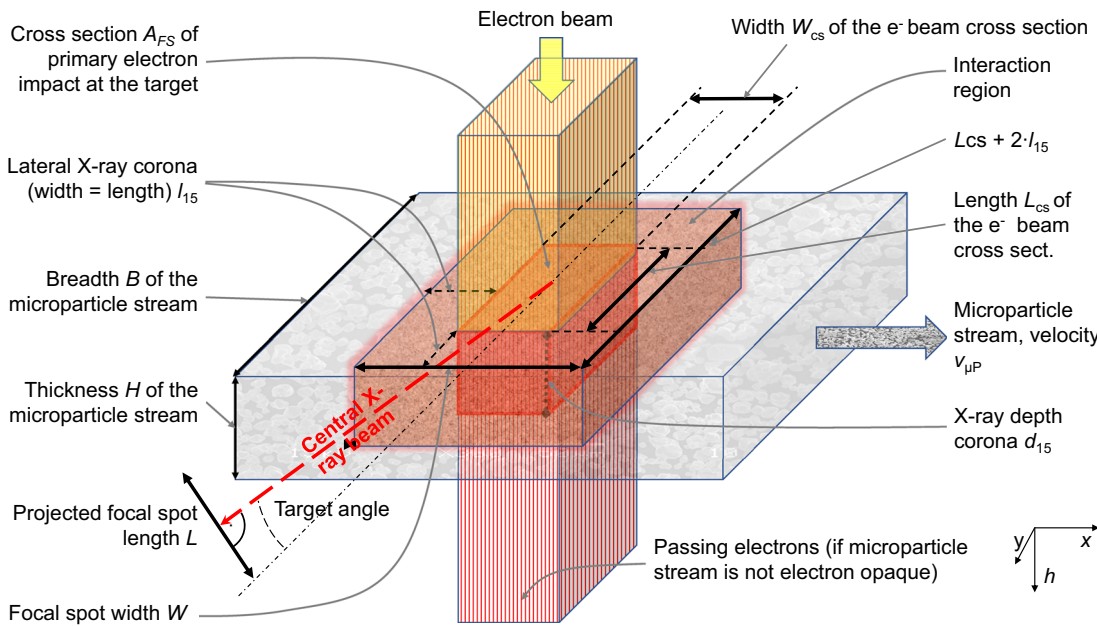

**Fig. 2 | Schematic of the interaction region of a microparticle stream with an orthogonal electron beam.** Potential X-ray corona margins from electron backscattering are indicated.

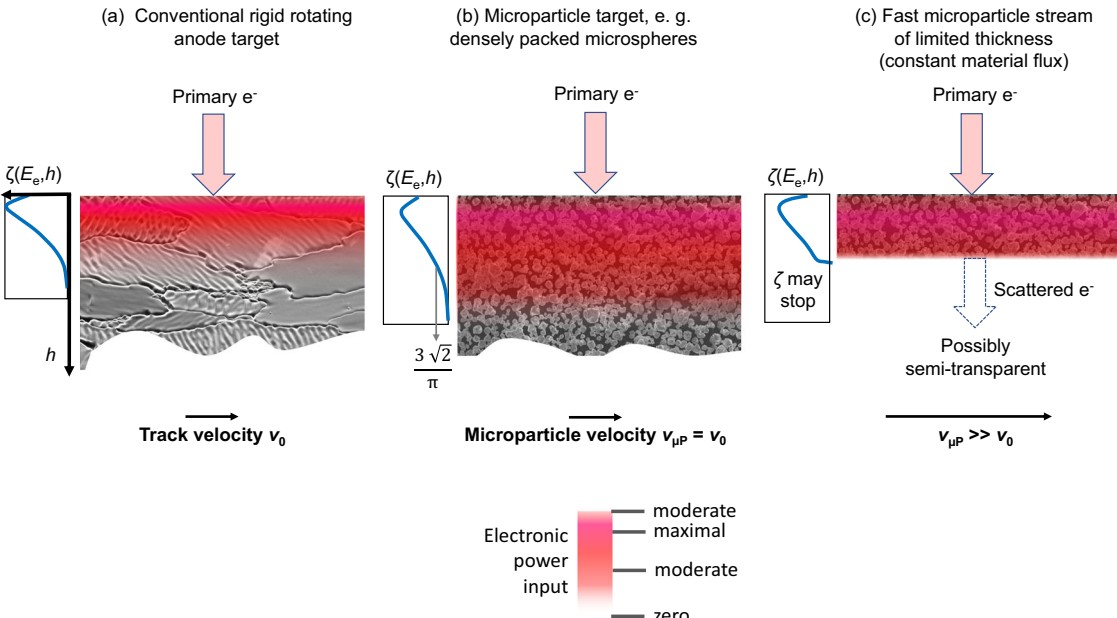

**Fig. 3 | Electron impact with energy $E_e$ on a compact and on microparticle targets.** The electronic power input distribution $\zeta(E_e, h)$, as a function of depth $h$ and energy $E_e$, is shown schematically for each case. A decreasing macroscopic mass density results in an extension of the region with the highest input power density (coloured band, colour code below). **a** An electron beam is impinging on a slowly moving focal track (velocity $v_0$). **b** The compact target may be replaced by a thick layer of hexagonal dense packed microspheres of constant flux, slowly moving also with $v_0$, and with reduced relative mass density of $\pi/(3\sqrt2)$. $\zeta$ stretches in $h$ by the inverse. **c** A further to $v_{\mu P}$ accelerated stream may be thinned by $v_0/v_{\mu P}$. For high $E_e$, $\zeta$ may stop upon partial electron transparency.

conversion efficiency may be compensated for by increased electron supply that modern cathodes would indeed be capable of if the gas pressure stays below $10^{-5}$ Pa[1]. Unconsumed energy may be harvested by electron collectors, that are known from Klystrons, or generate X-rays in an additional underlying target.

The transition from a compact to a microparticle target is schematically illustrated in Fig. 3. If a slowly moving rigid tungsten rotating anode moving with the velocity $v_0$, Fig. 3a, would be replaced by a sufficiently thick array of close-packed microspheres that move with the velocity $v_{\mu P}$, with $v_0 = v_{\mu P}$, Fig. 3b, the mass density would decrease by

$\pi/3\sqrt2$. With this modification, the distribution of the electronic input power $\zeta(E_e, h)$, that is a function of the depth, $h$, and the electron energy, $E_e$, would expand by the inverse of the density ratio. If the microparticles of constant flux were accelerated with to higher velocity $v_{\mu P}$, the macroscopic mass density would further decrease by $v_0/v_{\mu P}$. We illustrate this by keeping the stream dense but truncated, as illustrated by the thin layer in Fig. 3c. The stream may become semi-transparent for electrons and $\zeta$ may break off where the electrons exit. The residual power would be taken into the vacuum. In practical implementations, the stream may dissolve into a dilute cloud of separated microparticles with further

reduced density. The electron power loss function $\zeta$ would further expand in the depth direction (not shown for clarity).

In summary, the surface-heated eroding rigid anode is replaced by an electrically floating volume-heated stream of potentially much faster non-eroding microparticles with higher heat capacity due to a higher permitted temperature amplitude. The input power density can be enhanced. Most importantly, the size of the X-ray focal spot size may be reduced to the benefit of image sharpness.

## Results and discussion
### Neutral microparticle charge balance
Uncontrolled charging of microparticles by the electron beam should be avoided, as it may cause stream spreading, a false X-ray spectrum, and electron beam defocusing. As the spectral bandwidth and the X-ray intensity are defined by the difference between the electron emitter and target potential, the so-called tube voltage, any uncontrolled change of the electrical target potential would cause spectral distortion.

Monte Carlo (MC) simulations of electron transport for micrometre-sized tungsten spheres in free space suggest that the comparatively high electron backscatter yield of high atomic number tungsten helps to suppress charging for impact energies used in medical diagnostics, Fig. 4.

The backscattering yield as a function of the primary energy, Fig. 4a, b, even slightly exceeds unity above a well-defined energy threshold, Fig. 4c, that, beneficially, lies in the relevant range for diagnostic imaging. Below this energy, electrons cause negative charging, while causing slightly positive charging above. If operated at this neutrality threshold, the energy absorption amounts to about 11% for the investigated particle sizes. Figure 4d illustrates trajectories and the scattered electron energy for 150 keV sample primary electrons. The simulation reveals that the exit energy of the backscattered electrons and the backscattering yield both decrease with increasing microparticle size, Fig. 4e, approaching thick targets.

Dense streams of microparticles may exhibit more charging than is predicted from Fig. 4, as electrons may interact with multiple spheres with decreasing electron energy at each interaction. Initial simulations show that increasing the electron energy by 25 kV for 5 μm diameter spheres assumed to fly in a single monolayer would suffice to suppress charging. Thermionic electron emission at temperatures above about 1800 °C, photo-electric emission, or electron field emission from extremely small particles may also be employed. If necessary, the cathode voltage and electron focusing could be adapted.

### Tube power and X-ray output projection
Electron stopping in compact and granular targets can be treated identically for the purpose of this study since the material density can be reduced from the ratio of the heat capacity and the maximum value of the input power distribution shown in Fig. 5a, see Methods.

The gain of the permitted input power for tungsten microparticle streams can be compared quantitatively with a leading conventional CT rotating anode source, the iMRC® tube[1] (Philips, The Netherlands). As usual, this tube is specified using the Müller–Oosterkamp surface-heating model that ignores the tube voltage, see Supplementary Discussion. Volume-heating models such as that of Whitaker[30] and others have been suggested. However, lifetime tests have revealed that the performance is limited by focal track erosion and the Müller–Oosterkamp model is adequate. In contrast, a volumetric heating model applies to microparticle targets in the absence of inter-particle heat conduction. The maximum input power can be inferred from the power distribution function, $\zeta$, as determined by MC simulations. Figure 5a depicts the function $\zeta_{W}(E_e, h)$ for compact tungsten, maximum normed for the figure and shown for a selection of electron energies $E_e = -eU_{\text{tube}}$, where $U_{\text{tube}}$ denotes the voltage between electron emitter and target and $-e$ is the fundamental electron charge. The peak value $\zeta_{W\_peak}$ appears at

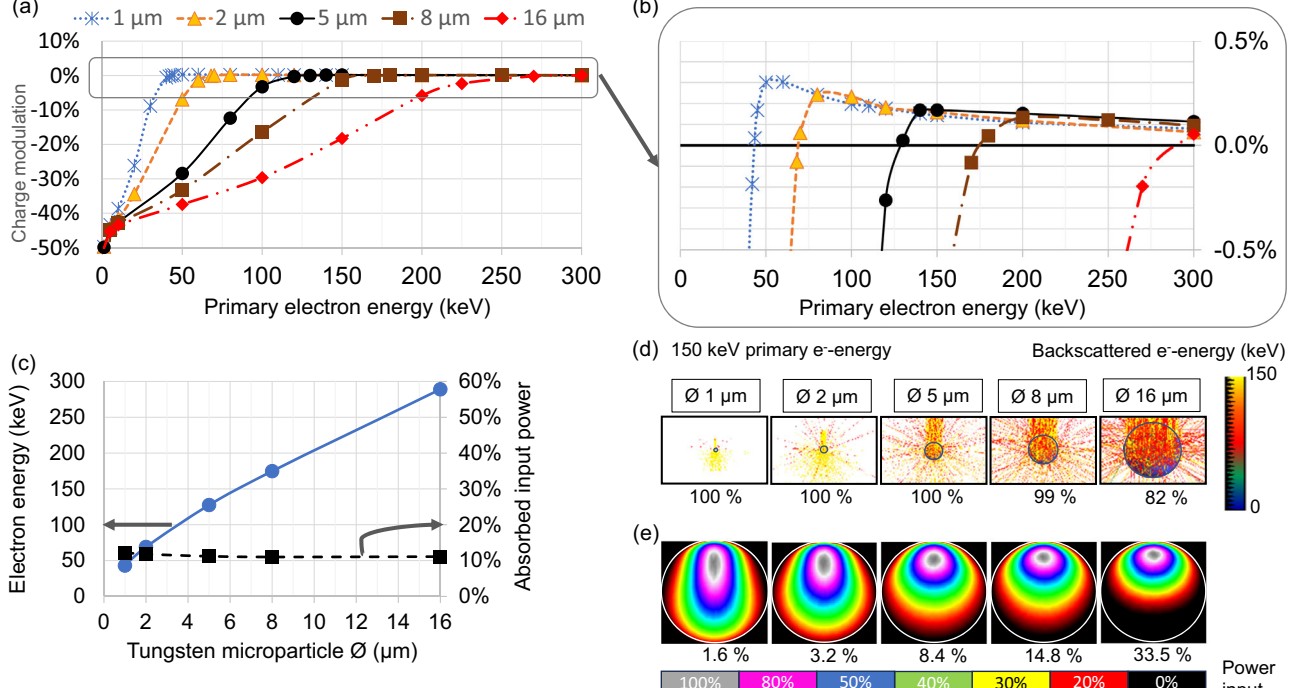

**Fig. 4 | Charging of microparticles under electron impact. a** MC of the percentage of charging of selected tungsten microspheres in free space for each impacting primary electron, $10^5$–$10^7$ primary events, max. variance 0.22%, typically 0.01%. **b** Zoom in on the vertical axis. **c** Primary threshold electron energy where charging vanishes (left scale) and percentage of absorbed energy for that threshold energy (right scale), as functions of the microsphere diameter. **d** Spectrum of expelled electrons (top part with colour code) and percentage of backscattered electrons for 150 keV primary energy (bottom row) for different microparticle sizes. **e** Stopping power distribution (total, integrated normal to centre plane, colour code: grey = max, black = zero) and percentage of absorbed energy (bottom row) in microparticles of different sizes for 150 keV primary energy.

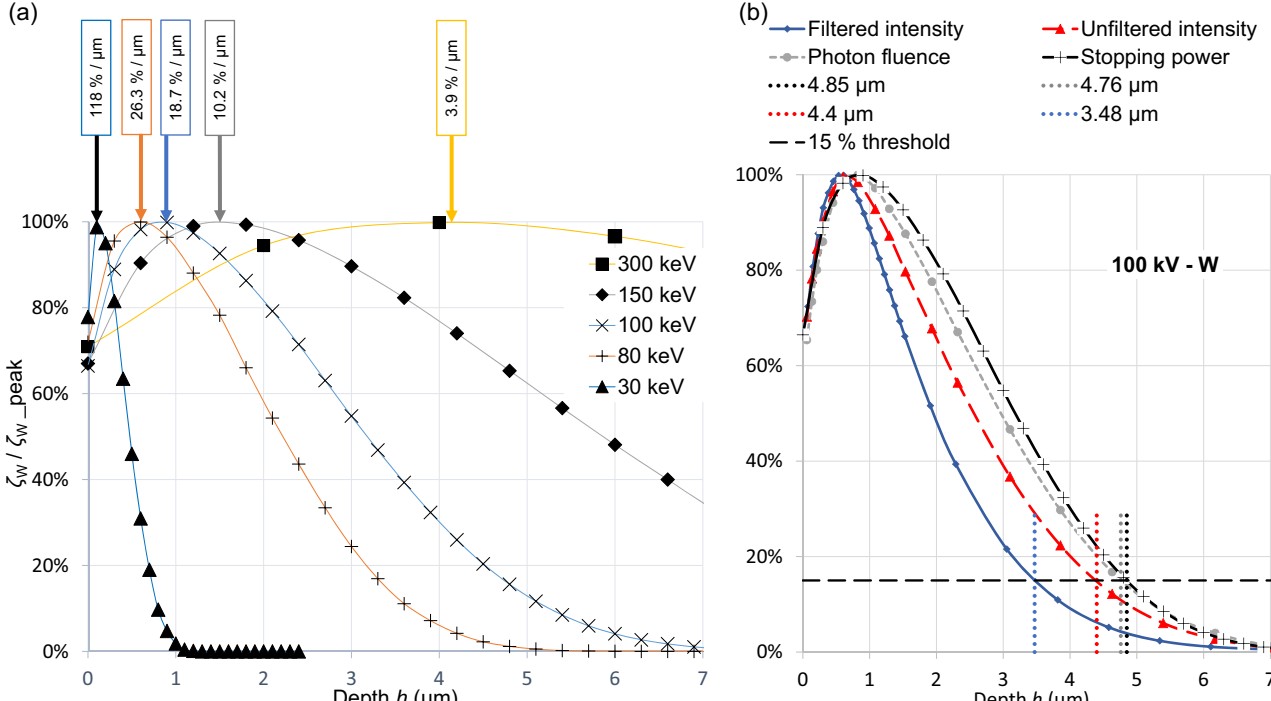

**Fig. 5 | Input power distributions for a compact target. a** MC of energy deposition profiles for compact tungsten $\zeta_W(E_e, h)$ vs. depth h for selected primary electron energies, $E_e = -e \cdot U_{tube}$, $U_{tube}$ being the tube voltage; $10^5$ primary events for 30 and 300 keV, $10^6$ for 80 and 150 keV, $5 \times 10^6$ for 100 keV. $\zeta_W(E_e, h)$ is shown normalised to its maximum $\zeta_W(E_e, d_{peak})$ that is listed for each energy in the boxes above. **b** Sample depth distributions of photon fluence and X-ray intensity compared with

the electronic power input distribution from the normal impact of 100 keV electrons as shown in (**a**). The extension of the corona of the radiation that is filtered by an additional 2.5 mm thick aluminium slab amounts to only 3.47 μm, compared with 4.85 μm. Therefore, the 15% depth, $d_{15}$, of the power distribution function $\zeta_W(E_e, h)$, is a conservative proxy for the X-ray corona.

the depth $h = d_{peak}$. In the absence of heat conduction, the local temporal temperature gradient $dT/dt$ at $d_{peak}$ is proportional to $p_{FS}\zeta_{W\_peak}$, where $p_{FS}$ denotes the primary electronic power density in the focal spot. This holds for compact as well as microparticle targets, assuming sufficient electron transparency of small enough particles, Fig. 4d, e, and Methods.

Figure 6a compares the development of classic X-ray tube technology with the potential that microparticle targets provide. It shows in the left panel the very limited gain of the specific relative focal spot power for rotating anode technology during the recent three decades. The performance has increased by only a few dozen percent for premium performance CT tubes based on the ratio of CT anode input power divided by $L_{CS}\sqrt{W_{CS}}$, according to the Oosterkamp-Müller model, and as a percentage of that value for the reference X-ray tube iMRC® (Philips, The Netherlands). Here $L_{CS}$ denotes the physical length and $W_{CS}$ the width of the focal spot as sketched in Fig. 2. The reference tube would enable only 25 kW nominal CT anode input power[32], as concluded from public power curves and assuming an extremely small focal spot for CT (0.3 nominal, 8° target angle). A giant leap can be expected from microparticle target technology. As microparticle targets interact with electrons like compact targets, the X-ray conversion efficiency of thick microparticle streams can be deemed equivalent. Thus, Fig. 6b represents the possible relative gain $G_{0.3}$ of permitted input power and the X-ray output according to Supplementary Notes. Microparticles are assumed to enter the interaction region at a temperature of 100 °C and exit at the melting point of tungsten. With a deviation of less than 5%, the point-wise simulated gain of permitted power input $G_{0.3}$ for a 0.3 nominal focal spot can be approximated by $G_{0.3} = 1.970 \times 10^{-5} v_{\mu P} U_{tube}^{1.4649}$, where the microparticle velocity, $v_{\mu P}$, is expressed in metres per second and the tube potential, $U_{tube}$, is given in kilovolts. As the gain increases with the tube voltage, CT and non-destructive testing applications benefit the most. The gain would further increase by at least 48% in case of permitted melting that should be followed

by in-flight re-solidification and, hence, would require additional cooling space.

## Microparticle flow management and acceleration
The typical intermittent operation of a medical CT system will allow for capturing microparticles after scanning and to refill a repository. However, details of the tungsten powder rheology in a vacuum and other details will be the subject of future publications.

If a classic rotating anode would be used for centrifugal acceleration, i.e., in a hybrid target configuration where microparticles may be filled into a concentric groove near the axis, spill out of radial channels near the focal spot and fly over the focal track, tangential particle velocities of 100 m s$^{-1}$ would become feasible. Magnetic rotors, as in turbo-molecular vacuum pumps, would allow for approximately 400 m s$^{-1}$ or more. Other than classic rotating anodes their rotors remain cool and at high yield strength without experiencing thermal stress or cycling. Thus, a magnetic rotor accelerator would be the solution of choice, and the corresponding performance gain $G_{0.3}$ is mapped in Fig. 6a (Microparticle target, for diagnostic CT). Due to particle neutrality, the magnetically levitated rotor would not necessarily carry current, which simplifies its design. Further electrostatic microparticle acceleration to velocities of the order of 1000 m s$^{-1}$ seems theoretically feasible[33] if neutralisation before concentrating the particles in the interaction region would be viable, such as by photo-electric emission under auxiliary UV radiation or other ionising means like a small gas load.

## Derating for dilute targets
In classical targets for diagnostic imaging, the dimensions of the focal spots are typically much larger than the average lateral electronic scattering range and penetration depth. It is then assumed that the focal spot width, $W$, equals $W_{CS}$, the width of the electron beam cross-section, and its X-ray optical length, $L$ (projected normal to the central X-ray beam), equals

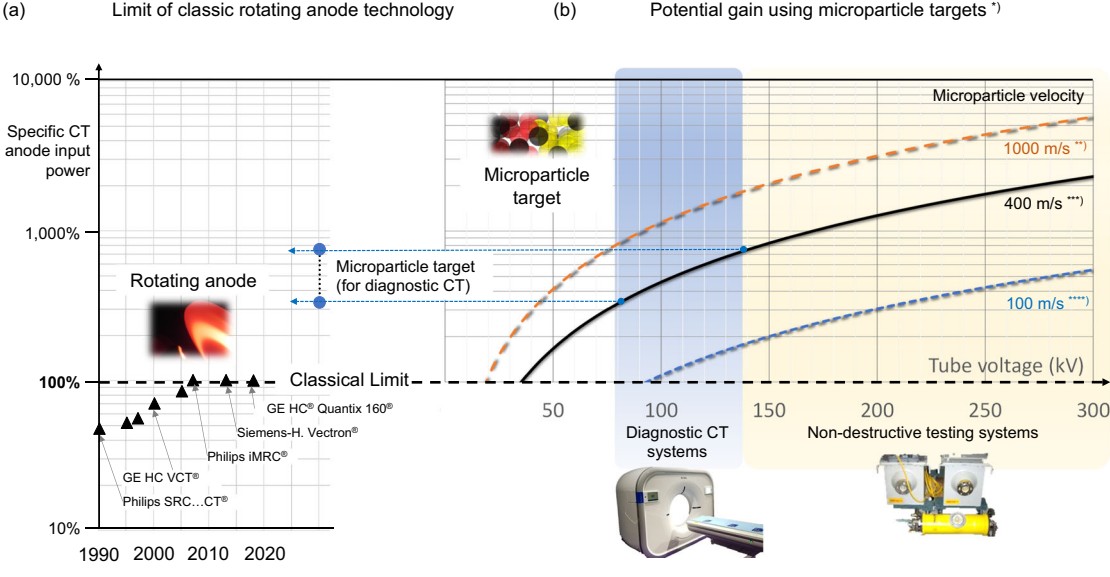

**Fig. 6 | Relative focal spot specific computed tomography (CT) anode input power (see text) of bremsstrahlung target concepts in relation to best-in-class rotating anode CT X-ray tubes. a** Historical development for premium commercial X-ray tubes. The improvement has stalled since the advent of the reference tube iMRC® (Philips, The Netherlands) in 2007 to about 25 kW for an assumed small CT focal spot (0.3 nominal, 8° target angle)[16]. **b** Given the historical limit (100 %), microparticle targets promise an unprecedented gain of performance that depends on tube voltage and microparticle velocities, as shown, assuming a close-packed,

sufficiently thick tungsten microparticle stream. Velocities $v_{\mu P}$, are marked. Tube voltage ranges for CT and for non-destructive testing are indicated. Reduction factors apply in case of lower microparticle density (see Supplementary Methods 1 and Supplementary Table for details). Footnotes: *) Focal spot 0.3 (nominal, 8° target); reference: Philips iMRC® tube; **) Enabled by combined mechanical and electrical acceleration; ***) Acceleration by a magnetically levitated high-speed rotor at ambient temperature; ****) Hybrid system: centrifugal microparticle acceleration in channels in a modified rotating anode.

$L_{CS} \cdot \tan(\alpha)$, where $L_{CS}$ is the length of the electron beam cross-section and $\alpha$ denotes the target angle, see Fig. 2. For reduced macroscopic mass density, $\rho_{\mu P}$, of a microparticle stream compared with a compact target of mass density, $\rho_W$, the electronic scattering range approximately scales with $\rho_W / \rho_{\mu P}$ if the discrete nature of microparticles can be ignored. A substantial lateral corona of X-ray production may exist with $W > W_{CS}$ and $L > L_{CS}$, and X-rays from within the microparticle stream may determine the length, $L$, for small anode angles. In such cases, the electron beam cross-section area $A_{FS}$, must be reduced to maintain a small X-ray emission region. For medical CT, the microparticle target mass density should exceed 1% of a compact target, see Supplementary Methods 1 and the Supplementary Table. The required material flow would have to be of the order of 260 g s$^{-1}$ to realise the focal spot referenced in Fig. 6, assuming 90% power absorption for 150 kV tube voltage and 400 m s$^{-1}$ particle velocity.

Figure 5b compares normalised X-ray corona and stopping power corona in relation to depth for 100 keV electrons impacting in tungsten. The 15% depth threshold, $d_{15}$, is adapted from the standard IEC 60336[16] upon the determination of focal spot dimensions from line-spread functions of X-ray intensity. The intensity for filtered radiation, $d_{15}$, from filtered intensity, was calculated using SpekPy v2.0[34], see Supplementary Methods 2. This refers to radiation in the central beam, filtered by the tungsten target and a 2.5 mm aluminium filter[35]. The result for 100 kV holds likewise for the other tube voltages. The wider stopping power margins compared with the X-ray coronae were taken as upper safety limits of the coronae ($d_{15}$, from stopping power) to calculate the derating coefficients in the Supplementary Table. Margins for lower mass density may be obtained by multiplication with $\rho_W / \rho_{\mu P}$.

An upper limit of $L$ can be approximated for small target angles, $\alpha$, by projecting the length of the electron beam cross-section, $L_{CS}$, of a rectangular classical focal spot of isotropic primary electron density onto the central beam axis, and by including the extensions $l_{15}$ and $d_{15}$ of the X-ray intensity corona in the lateral direction. $l_{15}$ denotes the lateral distance from the edge of an impinging electron beam, with assumed rectangular cross-section, to a position where the total X-ray intensity generated, at all depths, falls to 15% of the maximum value. If the stream thickness, $H$, is

not limiting, the projected X-ray focal spot dimensions become $L \approx \left(L_{CS} + 2l_{15}\right) \sin(\alpha) + d_{15} \cos(\alpha)$ and $W \approx W_{CS} + 2l_{15}$. To maintain classical dimensions and accommodate the coronae, $A_{FS}$, and, thus, the permitted input power must be reduced such that $A_{FS} = \left\{\left[L - d_{15}\cos(\alpha)\right] / \sin(\alpha) - 2l_{15}\right\}\left\{W - 2l_{15}\right\}$, see Supplementary Table.

### Improved image resolution

The spatiotemporal image resolution often limits the diagnostic efficacy, as shown in an illustrative sample coronary angiography image with iodine contrast in Fig. 7a, taken with a nominal focal spot of 1.0. Dotted arrows point to voxel arrays with a lack of small vessel contrast, in particular for fast-moving coronary arteries. Bold arrows indicate contrast blooming by partial volume artefacts that may cause false quantifications of stenoses when administering an iodine contrast agent.

The input power gain of a microparticle source translates to an improved modulation transfer function (MTF), Fig. 7b, bold line, compared with a classical high-performance rotating anode source referenced in Fig. 6b (horizontal dashed line). Figure 7b shows the MTF for the width direction on the vertical axis for exemplary focal spots assuming rectangular line-spread functions and with reference to the focal spot plane according to the standard IEC 60336[16]. A sample focal spot of standardised size 0.3 with 8° anode angle is taken for the microparticle source, as in Fig. 6b, that is operated for Fig. 7b with a sample tube voltage of 100 kV and assuming a particle velocity of 400 m s$^{-1}$. To allow for equivalent input power, the focal spot of a rotating anode tube must be enlarged, thus worsening its MTF and imaging performance. Patterns of resolvable line pairs in the inserts, that mimic images of line phantoms as objects, show qualitatively the difference of spatial resolution that depends on the tube voltage due to voltage-dependent benefits from volume heating of microparticle streams.

### Smallest achievable focal spot dimensions

X-ray coronae (Fig. 5 and Supplementary Methods Fig. 1) determine the achievable focal size in densely packed targets. If conversion efficiency can

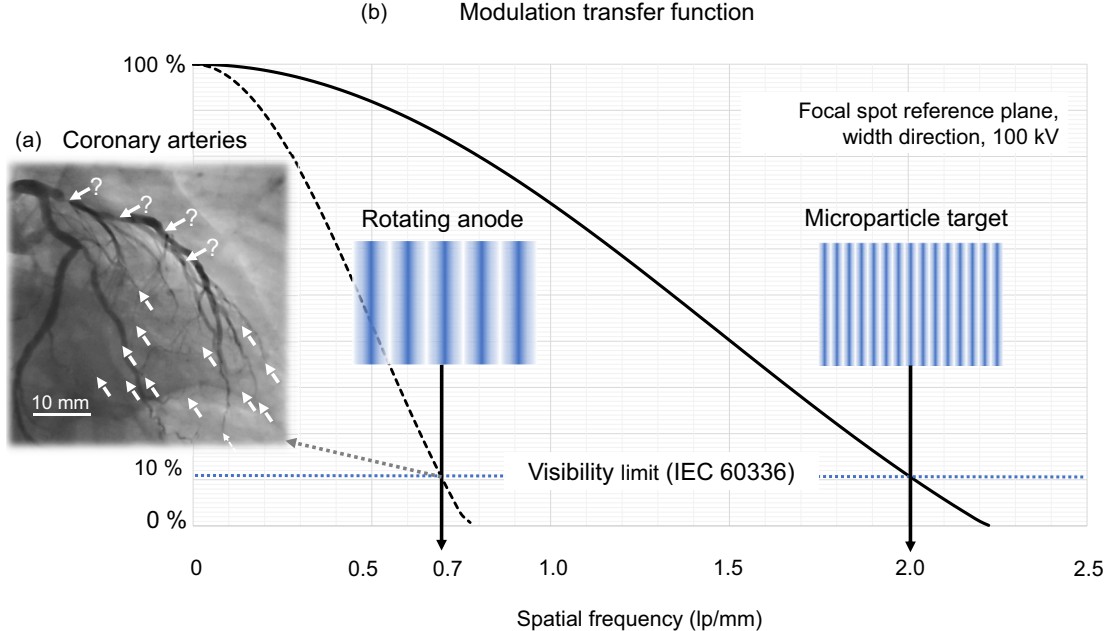

**Fig. 7 | Modulation transfer function[16] for the focal spot reference plane. a** Sample iodine contrasted coronary angiography of the human heart imaged with a Siemens-Healthcare Axiom-Artis® X-ray system (Megalix-Cat Plus® X-ray tube, nominal FS size 1.0, source-detector distance 80 cm, detector-patient distance 20 cm, pulse width 6.8 ms, tube voltage 84 kV, tube current 756 mA). The limited spatiotemporal resolution causes partial volume artefacts (bold arrows with question marks) and poor or non-existent delineation of small arteries (dotted arrows). **b** Comparison of modulation transfer functions for the focal spot width (vertical axis) of a rotating anode tube (dashed line) with a microparticle source of equivalent total input power (bold line). The spatial frequency is given in line pairs per millimetre (lp/mm) assuming an isotropic tube current density, a rectangular line-spread function, a sample focal spot of 0.3/8° (nominal), and a tube voltage of 100 kV. To allow for equivalent input power and depending on tube voltage, inevitably a larger focal spot is necessary for a rotating anode tube, which significantly impairs its modulation transfer function. Resolvable line patterns show qualitatively the gain of spatial resolution. Inserts: Normal line patterns mimicking the images of line phantoms comprising X-ray opaque linear absorption stripes with pitch 0.7 lp/mm (left) and 2.0 lp/mm (right).

be sacrificed, the minimal achievable focal spot length approaches the microparticle diameter. The minimal width equals the achievable width of the electron beam that may approach a few dozen micrometres in practice.

## Multispectral imaging and focal spot deflection

Larger microparticles produce an enhanced percentage of low-energy photons from low-energy electrons that tend to be absorbed with a higher ratio instead of being backscattered. Hence, microparticle streams of varying particle size or density may support multispectral imaging by producing photons of varying mean energies. Due to advances in filtering, thin-target-type microparticle streams may become excellent sources of nearly monochromatic X-rays with high intensity.

As with classical anode technology, focal spot deflection would be possible, too, by deflecting the electron beam.

Using multiple streams of microparticles in various directions may offer additional options.

## Methods

The intensity of filtered X-radiation was calculated using the tool kit SpekPy v2.0[34], see Supplementary Methods 2. The MC code Casino v. 3.3.0.4[36] with parameter settings stated in Supplementary Methods 3 was used throughout this study and, in addition to other authors, validated as described in Supplementary Methods 4. Charging of single spherical microparticles with an electron beam of isotropic current density was investigated as the number difference between primary impacting and backscattered electrons. The percentage of absorbed energy per microparticle was extracted from the ratio of the volume-integrated absorbed energy and the number of input trajectories times the primary energy per electron.

The depth distribution of the electronic power input for microparticle and compact targets was simulated to assess the local maximum of the input of electronic power into the microparticle stream. Electron stopping in compact as well as in microparticle targets can be treated

likewise for the purpose of this study, as results scale in a simple manner with material density. Provided, inter-particle electric fields are absent and effects on atomic scales can be ignored, impacting electrons change their energy state only by nuclear or electronic scattering at atoms, not in free flight through the vacuum. The temperature distribution of an assembly of sufficiently small microparticles in a microparticle stream under electron impact can simply be inferred from the temperature distribution in a compact tungsten target, whereby heat conduction is ignored. Let the surface density of input power supplied to a target of assumed infinite thickness be $p_{FS} = U_{tube} \cdot j_{FS}$. Here, $U_{tube}$ represents the tube voltage, and $j_{FS}$ the density of input current (without subtracting the backscattered electron current). The surface density of absorbed power, integrated over the entire depth $h$, is then $p_{FS} (1 - \eta_{energy})$, with $\eta_{energy}$ denoting the energy backscattering yield. Further, let the kinetic input electron energy $E_e$ and depth-dependent function $\zeta$ be the percentage of electronic stopping power in a volume, divided by its infinitesimal thickness $dh$ at $h$ and the input power density $p_{FS}$; $\zeta(E_e, h)$, respectively $\zeta_W(E_e, h)$ for tungsten. The assessment of power deposition and permissible material-specific maximal input power simplifies to a one-dimensional problem, if the cross-section of the electron beam by far exceeds the lateral scattering range of electrons in all directions, as for compact targets in typical medical imaging. Stopping power from absorbed electrons is deposited in the depth according to $\int_0^\infty \zeta(E_e, h) dh = 1 - \eta_{energy}$.

To assess $\zeta$ by MC, energy deposition was simulated by injecting electrons of energy $-e \cdot U_{tube}$ into the circular focal spot of area $A$ that the tool Casino requires. To avoid errors from lateral scattering, the diameter of the impacted area $A$ is chosen substantially larger than the stopping range of electrons in tungsten in the continuous slowing down approximation (CSDA). To further minimise the errors, only a restricted volume of voxels underneath a rectangle with reduced surface area $A_{FS}$, hit by $n = n' (A_{FS}/A)$ electrons, is evaluated; n relates to the current density and exposure time $\Delta t$

by $n = -A_{FS} \cdot (j_{FS}/e) \cdot \Delta t$ and, thus, $A_{FS} \cdot j_{FS} \cdot \Delta t = -e \cdot n$. The energy supply $E_{FS}$ during the time $\Delta t$ is simulated according to $E_{FS} = A_{FS} \cdot U_{tube} \cdot j_{FS} \cdot \Delta t = -e \cdot n \cdot U_{tube}$. The MC code sums the stopping energy per target voxel in the evaluated volume. Let $\Delta E_i$ be the summed energy for all voxels in a slab of cross-section $A_{FS}$ and thickness $\Delta h$, stacked down the depth $h$-axis and indexed by $i$. $\zeta$ may then be numerically approximated as $\zeta (i\Delta h) \approx (\Delta E_i/\Delta h)/E_{FS}$. $\zeta$ peaks in tungsten as $\zeta_{W\_peak}$ at $d_{peak}$, at about between 8% ($E_e = 30\,kV$) and 6% ($E_e = 300\,kV$) of the tabulated CSDA electron stopping range. The simulation outputs the numbers of backscattered electrons and the amount of energy backscattering $\eta_{energy} = 1 - [(-e) \cdot n \cdot U_{tube}]/\sum_i E_i$.

## Reporting summary

Further information on research design is available in the Nature Portfolio Reporting Summary linked to this article.

## Data availability

The authors declare that the findings of this study are available within the paper and its Supplementary Information. The datasets generated during and/or analysed in this study are available from the corresponding author upon reasonable request.

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

## Acknowledgements

The authors thank the reviewers and the editorial team for their thorough analysis and valuable suggestions for the improvement of the draft article. No external funding was used.

## Author contributions

R.B. developed the concept, simulated, and authored the paper. C.H. advised on the rheology of microparticles and materials science. G.P. simulated the output and depth profile of produced X-rays. P.T. provided benchmarking data, linked to microparticle physics in fusion reactors, and wrote the supplementary material text in cooperation with the main author. M.D. initiated the study and linked it to medical physics application. R.B. prepared the original draft, and all authors critically reviewed and approved the manuscript.

## Funding

## Competing interests
M.D. is the owner of Innovicum, Sweden. The company is the assignee of a relevant patent application. R.B. is the owner of XtraninX, Technical Consulting (www.XtraninX.com) and named as inventor for a relevant patent application. All other authors declare no competing interests.

## Ethics approval
The research has included local researchers throughout the research process.
