## [Peer Review File · Communications Engineering]

REVIEWERS' COMMENTS:

Reviewer #1 (Remarks to the Author):

The authors present a theoretical study of a new concept of X-ray sources. This is based on the utilization of linearly moved microparticles instead of the commonly used solid rotating target, which still shows the limits despite the effort to improve. The introduced idea is supported by several simulations that are mainly focused on state-of-the-art X-ray tube parameters of a medical CT system.

The presented idea is very interesting and innovative. However, the whole manuscript is written in an unsystematic way where it is hard to follow individual steps that serve as the first theoretical proof of concept. It is difficult to stay oriented in various parts of the manuscript because nobody is informed in advance about what should be tested or solved. It is also partially because there is a large number of references to extended and supplementary data. So, for example, the result part will start with solving a charge balance of microparticles and the reader is not informed about the importance, purpose or expected outcome. I would recommend to familiarize the reader with what is necessary to find out the design of such a target and what will be solved specifically in this manuscript. This probably belongs to the introduction, which is more like motivational words now than the research of current knowledge and what must be done to prove the new ideology.

Here are specific comments in accordance with this statement above:

- Even though the authors refer to standards about terminology, it would be better for readers to get an understanding of it as the standards are not openly accessible. Please explain nominal focus spot size (compared to optical, physical or effective focal spot), CT anode input power, and gain of permitted power input power.

- Is the tilt of the electron beam to the stream of microparticles under some angle, too? Why not show it in Figure 2?

- Methodology for simulations of graphs in Figure 3 is not known. Explain more Fig 3(c) It is hard to understand the functions and where the threshold comes from, as well as the meaning of colours.

Line 117-118 the idea of hybrid configuration is not clear. Does it mean microparticles and a solid target below that?

- Hexional particles (line 124) and tungsten spherical particles (line 137) are mentioned. Which one is actually the chosen geometry and why?

Figure 4 needs more attention to be commented on. It is not clear if the axis of tube voltage is somehow connected with data from “last years”, what are the units of CT anode input power if the maximum is 1 kW for rotating anode?

Line 191 what is the reference tube?

Figure 5a How this image was made? There is missing methodology (system settings, sample preparation, voxel resolution, ..) It is really questionable if the image is wrong or staining or sample and so on. Please consider if this image is necessary to show in such a theoretical/simulating type of paper if there is no image with improved MTF. It could rather be used in the introduction to support the motivation.

Line 273 can you explain blooming artefacts? I suppose this is not any known CT artefact.

Line 320 There are mentioned rotors with angular velocities; this is inconsistent with the manuscript which promises linear motion of microparticles.

- Does temperature play a role only in discharging particles, or does it also play a big role in mechanical stability and X-ray generation?

- The authors came up with separate outputs like

Line 246 “the microparticle target mass density should exceed 1%“

Line 153 “except for the 16um”

Line 214-219 velocity

Would it be possible to summarize them into one optimal target design (meaning density, particle size, velocity) and see it in 3D rendered image + generate what focal spot size on maximum power?

Reviewer #2 (Remarks to the Author):

99 operation have been identified that allow management of charge build-up from electron
100 bombardment by backscattering and other discharging mechanisms. This ensures a well
defined X-ray spectrum

- Some words should be added to explain this claim.

143 (d) Spectrum of expelled electrons (top row) and percentage of absorbed electrons for 150 keV
primary energy (bottom row)

- May be top part and right-side part is better description.

305 Dilute microparticle streams may support multispectral imaging by varying the microparticle
306 size and density and, therefore, producing photons of varying mean energies.

- The same as in case of rows 100 and 101 (at least basic explanation should be added).

317 X-ray output is independent of the focal spot width;

- is this statement valid?

322 multispectral imaging may be supported by varying the microparticle size and density

- the same as in case of rows 100 and 101 (at least basic explanation should be added).

Reviewer #3 (Remarks to the Author):

This article is concerning high-brilliant X-ray source with new concept, which is adapting high-speed micrometer size particles for the target materials instead of rotating disc target. The conceptual idea is very interesting, if it can be realized. However, I could not imagine the reality of this concept without the clear idea to overcome the challenges as authors also mentioned from line 200 to 203 in the manuscript. The authors mentioned “this will be discussed in future

publication". The authors suggestions to overcome the challenges of the technical and/or theoretical details is undoubtedly essential for the publication; how to accelerate high-density (at least 1% of bulk) particle stream, how many particles or weight of them is required, how to receive high-temperature high-speed particles (and recycling?), and so forth.

The authors thank the referees for the positive evaluation of our work and for the valuable remarks. Please find our answers and improvements listed below.

Reviewer #1

Chapter, line, Fig.	Reviewer comment	Author reply, author correction	Chapter, line, fig. of improved manuscript
Intro- duction	The authors present a theoretical study of a new concept of X-ray sources. This is based on the utilization of linearly moved microparticles instead of the commonly used solid rotating target, which still shows the limits despite the effort to improve. The introduced idea is supported by several simulations that are mainly focused on state-of-the-art X-ray tube parameters of a medical CT system. The presented idea is very interesting and innovative. However, the whole manuscript is written in an unsystematic way where it is hard to follow individual steps that serve as the first theoretical proof of concept. It is difficult to stay oriented in various parts of the manuscript because nobody is informed in advance about what should be tested or solved. It is also partially because there is a large number of references to extended and supplementary data. So, for example, the result part will start with solving a charge balance of microparticles and the reader is not informed about the importance, purpose or expected outcome. I would recommend to familiarize the reader with what is necessary to find out the design of such a target and what will be solved specifically in this manuscript. This probably belongs to the introduction, which is more like motivational words now than the research of current knowledge and what must be done to prove the new ideology.	The Introduction has been substantially rephrased, guidance and explanatory text added throughout the paper and text moved to enhance clarity. We added more technical details, such as mechanical acceleration of microparticles, in the Introduction and the required material flow under Results and Discussion Challenges have been ranked and the most critical challenge (charging) is considered in detail. We have added remarks on how to remediate excessive charging for other conditions than simulated. Other technical challenges have been listed and concepts of resolution outlined in the additional text. Design parameters, have been added and the suggested microparticle density has been considered.	Intro- duction 59-71, Results and Discussion, 240-241
	Here are specific comments in accordance with this statement above:  - Even though the authors refer to standards about terminology, it would be better for readers to get an understanding of it as the standards are not openly accessible. Please explain nominal focus spot size (compared to optical, physical or effective focal spot), CT anode input power and gain of permitted power input power. 	We explicitly translated the standards terminology to physical terms in the introduction.	46-50, 199-205

Fig. 2	- Is the tilt of the electron beam to the stream of microparticles under some angle, too? Why not show it in Figure 2	We have explicitly stated normal impact.	Fig. 2, 167
Fig. 3	- Methodology for simulations of graphs in Figure 3 is not known. Explain more Fig 3(c) It is hard to understand the functions and where the threshold comes from, as well as the meaning of colours.	We have added descriptive text and improved the meaning of the energy threshold. The color code of Fig. 3e (now Fig. 4e in the updated version) is stated verbally. A Min/max description of the color code in Fig. 4e (arrows) was added	Fig. 4, 141-146
117-118	the idea of hybrid configuration is not clear. Does it mean microparticles and a solid target below that?	We have added an explanation.	100, 214
124, 137	- Hexonal particles (line 124) and tungsten spherical particles (line 137) are mentioned. Which one is actually the chosen geometry and why?	We have rephrased the paragraph to improve clarity.	237-238
Fig. 4	Figure 4 needs more attention to be commented on. It is not clear if the axis of tube voltage is somehow connected with data from "last years", what are the units of CT anode input power if the maximum is 1 kW for rotating anode?	We have improved the graph for clarity and turned the vertical axis into a percentage scale for clarity. We have also improved the caption.	Fig. 6, 187
191	Line 191 what is the reference tube?	We have specified the reference tube.	199-201
Fig. 5a	Figure 5a How this image was made? There is missing methodology (system settings, sample preparation, voxel resolution, ..) It is really questionable if the image is wrong or staining or sample and so on. Please consider if this image is necessary to show in such a theoretical/simulating type of paper if there is no image with improved MTF. It could rather be used in the introduction to support the motivation.	We have explicitly specified the image as a sample image and added (in the main text and in the caption) the nominal value of the focal spot used for the depicted cine frame.	265-266, Fig. 7, 283
273	Line 273 can you explain blooming artefacts? I suppose this is not any known CT artefact.	We have added the, in this case, synonymous term partial volume artifact.	267-268, Fig. 7, 283
320	Line 320 There are mentioned rotors with angular velocities; this is inconsistent with the manuscript which promises linear motion of microparticles.	We have integrated the paragraph comprising this bullet point into the Introduction and added text explaining the function of a rotating member.	60-62
	- Does temperature play a role only in discharging particles, or does it also play a big role in mechanical stability and X-ray generation?	X-ray generation is agnostic of temperature if the relatively small thermal expansion is ignored. It is critical for mechanical stability, however. Therefore cooling, slowing down and capturing of microparticles will be a special engineering task as described in more detail in the Introduction	64-65
256, 153, 214-216	- The authors came up with separate outputs like Line 246 "the microparticle target mass density should exceed 1%" Line 153 "except for the 16um"	Being the basis of future developments, the present work is of conceptual nature. It describes the paradigm shift of using	59-71, 237-242

	Line 214-219 velocity Would it be possible to summarize them into one optimal target design (meaning density, particle size, velocity) and see it in 3D rendered image + generate what focal spot size on maximum power?	independent microparticles instead of anodes that are conductively coupled to a power terminal. The details of the technical realization will be the subject of future work. We added more details, however, in the section Derating for dilute targets and in the Introduction.	
--	---	--	--

Reviewer #2

Chapter, line, Fig.	Reviewer comment	Author reply, author correction	Chapter, line, fig. of improved manuscript
99	99 operation have been identified that allow management of charge build-up from electron 100 bombardment by backscattering and other discharging mechanisms. This ensures a well defined X-ray spectrum	We have added a description.	128-131
Fig.4, 143	143 (d) Spectrum of expelled electrons (top row) and percentage of absorbed electrons for 150 keV primary energy (bottom row) - May be top part and right-side part is better description.	The caption has been improved.	Fig. 4d, 137-138
100-101, 305-306	305 Dilute microparticle streams may support multispectral imaging by varying the microparticle 306 size and density and, therefore, producing photons of varying mean energies. - The same as in case of rows 100 and 101 (at least basic explanation should be added).	We have added a detailed description.	296-300
317	317 X-ray output is independent of the focal spot width; - is this statement valid?	Yes, it holds for all volume heated targets where the thermal diffusivity is small or zero, as for microparticle based targets or circumstances where volume heating dominates. For rotating anodes, the dependency on the focal spot is predicted by the alternative model of surface heating that is suitable for rigid classical rotating anodes but not valid for the microparticle target that is volume heated.	
322	322 multispectral imaging may be supported by varying the microparticle size and density - the same as in case of rows 100 and 101	We have added a detailed description.	296-300

Reviewer #3

Chapter, line, Fig.	Reviewer – Comment	Author reply, author correction	Chapter, line, fig. of improved manuscript
200-203	This article is concerning high-brilliant X-ray source with new concept, which is adapting high-speed micrometer size particles for the target materials instead of rotating disc target. The conceptual idea is very interesting, if it can be realized. However, I could not imagine the reality of this concept without the clear idea to overcome the challenges as authors also mentioned from line 200 to 203 in the manuscript. The authors mentioned “this will be discussed in future publication”. The authors suggestions to overcome the challenges of the technical and/or theoretical details is undoubtedly essential for the publication; how to accelerate high-density (at least 1% of bulk) particle stream, how may particles or weight of them is required, how to receive high-temperature high-speed particles (and recycling?), and so forth.	We added more technical details, such as mechanical acceleration of microparticles, in the Introduction and the required material flow under Results and Discussion Challenges have been ranked and the most critical challenge (charging) is considered in detail. We have added remarks on how to remediate excessive charging for other conditions than simulated. Other technical challenges have been listed and concepts of resolution outlined in the additional text. Design parameters, have been added and the suggested microparticle density has been considered.	Introduction 59-71, Results and Discussion, 240-241

REVIEWERS' COMMENTS:

Reviewer #2 (Remarks to the Author):

Original remarks to the Author:

99 operation have been identified that allow management of charge build-up from electron
100 bombardment by backscattering and other discharging mechanisms. This ensures a well
defined X-ray spectrum

- Some words should be added to explain this claim.

143 (d) Spectrum of expelled electrons (top row) and percentage of absorbed electrons for 150 keV
primary energy (bottom row)

- May be top part and right-side part is better description.

305 Dilute microparticle streams may support multispectral imaging by varying the microparticle
306 size and density and, therefore, producing photons of varying mean energies.

- The same as in case of rows 100 and 101 (at least basic explanation should be added).

317 X-ray output is independent of the focal spot width;

- is this statement valid?

322 multispectral imaging may be supported by varying the microparticle size and density

- the same as in case of rows 100 and 101 (at least basic explanation should be added).

After corrections remarks to the Author:

The reviewer has no more remarks.

Reviewer #3 (Remarks to the Author):

Referee Report

This article is concerning high-brilliant X-ray source with new concept, which is adapting high-speed micrometer size particles for the target materials instead of rotating disc target. The conceptual idea is very interesting, if it can be realized. However, I could not imagine the reality of this concept without the clear idea to overcome the challenges as authors also mentioned from line 200 to 203 in the manuscript. The authors mentioned “this will be discussed in future publication”. The authors suggestions to overcome the challenges of the technical and/or theoretical details is undoubtedly essential for the publication; how to accelerate high-density (at least 1% of bulk) particle stream, how many particles or weight of them is required, how to receive high-temperature high-speed particles (and recycling?), and so forth.

The basic idea of this paper is very interesting, and a lot of numerical simulations were carried out.

However, there are no realistic proposals to create very fast high-density particles flow and I can't recognize any practical solutions.

Rebuttal letter,

Status July 10th, 2024

The authors thank the referees for the positive evaluation of our work and for the valuable remarks. Please find our answers and improvements listed below.

Reviewer #2

Chapter, line, Fig.	Reviewer comment	Author reply, author correction	Chapter, line, fig. of improved manuscript
	After corrections remarks to the Author: The reviewer has no more remarks.		

Reviewer #3

Chapter, line, Fig.	Reviewer – Comment	Author reply, author correction	Chapter, line, fig. of improved manuscript
	The basic idea of this paper is very interesting, and a lot of numerical simulations were carried out. However, there are no realistic proposals to create very fast high-density particles flow and I can't recognize any practical solutions.	We added the description of a repository of microparticles that can be replenished by gravity after each medical computed tomography scan when the tube, mounted in the rotary gantry of a computed tomography system, stops in 12 o'clock idle position. We added text on rotors with high circumferential velocity that have been in common use for UHV turbo-molecular pumps and that are suitable candidates for “powder pumps” to accelerate microparticles in ultra-high vacuum. Other than for the use as rotating X-ray anodes, that has indeed been tried before and that would require a current feed device, magnetic borne rotors used as microparticle pumps will not need to carry electric current. This aspect may become	Line 258 Lines 260-269

		important for technical implementations. We further added mentioning the concept of a hybrid system, i.e. a combination of classic rotary anode equipped with radial channels as microparticle accelerators and stated the achievable velocities. Very early initial experimental results suggest the rheologic suitability of sufficiently thin and wide channels for centrifugal acceleration. We additionally discuss how to overcome some extent of negative charging of dense microparticle streams or streams with larger microparticles treating the exemplary of a dense monolayer of 5 μm microparticles from premature results that have recently become available. We added text on discharging methods to avoid negative charging. We added text on the electric or mechanical separation of the cathode space comprising the high-voltage insulator, and the target space with microparticles, to avoid high-voltage instabilities.	Lines 232 (Fig. 6) and 260-263 Lines 185-189 Lines 96-101
--	--	--	--

Editor remarks

Chapter, line, Fig.	Editors comment	Author reply, author correction	Chapter, line, fig. of improved manuscript
New title	The editors recommend the following title: A compact X-ray source via fast microparticle streams	Title changed accordingly	Title

Abstract	The abstract advertises your paper and ideally appeals to a broad audience. You will need to revise your abstract as it does not conform to our house style. Please revise following the structure outlined below. Please make it as accessible as possible and avoid or explain specialist terms. ...a Please also remove any claims of primacy or hyperbolic statements	We streamlined the abstract and adjusted it to the recommended 150 words. We rephrased primacy or hyperbolic statements	Abstract
Web	Rolf Behling and colleagues propose a new X-ray source concept by replacing the rotating body with a linear stream of microparticles in the electron beam. The concept could make for smaller X-ray sources for computed tomography, radiation cancer therapy and non-destructive testing.	We recommend: Rolf Behling and colleagues propose a new X-ray source concept for high-resolution X-ray computed tomography and non-destructive testing as well as improved efficacy of radiation cancer therapy by replacing the rotary anode with a fast stream of microparticles in the electron beam.	Web
All figures	Use of Arial font	Complied with throughout	all
Typeset for math. terms	Italics vs Roman	Complied with throughout	All
Fig. 7a	Microscopy images and photographs in each Figure and Please add scale bars and corresponding definitions to Fig 7a.	Scale bar added. Definitions are in the text	Fig. 7a
Data Availability Statement	Please ensure that your Data Availability Statement complies with our Data Availability policy.	Complied with	End of the text
Supplementary Information	Any text in the Supplementary Information must be labelled with one of the following subheadings: Supplementary Notes (1,2, 3...), Supplementary Methods or Supplementary Discussion.	Complied with	Supplementary Information
Further improvement		In further response to the initial critics of reviewer 1 (in the first round of reviewing), we further improved the coherence and logic of the text, streamlined it, improved	Whole text

		the language, and sharpened some descriptions by using dedicated terminology.	
--	--	---	--